# Holey Carbon Nanohorns-Based Nanohybrid as Sensing Layer for Resistive Ethanol Sensor

**DOI:** 10.3390/s25051299

**Published:** 2025-02-20

**Authors:** Bogdan-Catalin Serban, Niculae Dumbravescu, Octavian Buiu, Marius Bumbac, Mihai Brezeanu, Cristina Pachiu, Cristina-Mihaela Nicolescu, Oana Brancoveanu, Cornel Cobianu

**Affiliations:** 1National Institute for Research and Development in Microtechnologies, IMT-Bucharest, 126 A Str. Erou Iancu Nicolae, 077190 Voluntari, Romania; bogdan.serban@imt.ro (B.-C.S.); niculae.dumbravescu@imt.ro (N.D.); octavian.buiu@imt.ro (O.B.); cristina.pachiu@imt.ro (C.P.); oana.brincoveanu@imt.ro (O.B.); 2Sciences and Advanced Technologies Department, Faculty of Sciences and Arts, Valahia University of Târgoviște, Aleea Sinaia, nr 13, 130004 Târgoviște, Romania; 3Institute of Multidisciplinary Research for Science Technology, Valahia University of Târgoviște, Aleea Sinaia, nr 13, 130004 Târgoviște, Romania; cristina.nicolescu@valahia.ro; 4Faculty of Electronics, Telecommunications and IT, National University of Science and Technology Politehnica Bucharest, Bd Iuliu Maniu 1-3, 061071 Bucharest, Romania; mbrezeanu@upb.ro; 5Academy of Romanian Scientists (ARS), Str. Ilfov Nr. 3, Sector 5, 050044 Bucharest, Romania; cornel.cobianu@gmail.com; 6eBio-Hub Center of Excellence in Bioengineering, National University of Science and Technology Politehnica Bucharest, Blvd. Iuliu Maniu Nr. 6, Sector 6, 061344 Bucharest, Romania

**Keywords:** ethanol sensor, holey carbon nanohorns, graphene oxide, swelling, HSAB principle

## Abstract

The study presents the ethanol vapor sensing performance of a resistive sensor that utilizes a quaternary nanohybrid sensing layer composed of holey carbon nanohorns (CNHox), graphene oxide (GO), SnO_2_, and polyvinylpyrrolidone (PVP) in an equal mass ratio of 1:1:1:1 (*w*/*w*/*w*/*w*). The sensing device includes a flexible polyimide substrate and interdigital transducer (IDT)-like electrodes. The sensing film is deposited by drop-casting on the sensing structure. The morphology and composition of the sensitive film are analyzed using scanning electron microscopy (SEM), Energy Dispersive X-ray (EDX) Spectroscopy, and Raman spectroscopy. The manufactured resistive device presents good sensitivity to concentrations of alcohol vapors varying in the range of 0.008–0.16 mg/cm^3^. The resistance of the proposed sensing structure increases over the entire range of measured ethanol concentration. Different types of sensing mechanisms are recognized. The decrease in the hole concentration in CNHox, GO, and CNHox due to the interaction with ethanol vapors, which act as electron donors, and the swelling of the PVP are plausible and seem to be the prevalent sensing pathway. The hard–soft acid-base (HSAB) principle strengthens our analysis.

## 1. Introduction

Ethyl alcohol is a volatile, flammable, colorless liquid with a typical wine-like aroma and pungent taste [1]. It is a common industrial raw material encountered in many chemical industrial processes and consumer products [2]. Thus, ethanol is extensively used in medicine (manufacturing processes of pharmaceutical preparations such as lotions, tonics, rubbing compounds, as well as antidote for both methanol and ethylene glycol poisoning) [3,4], cosmetics (mouthwash products, soaps, perfumes) [5], chemical industry (excellent solvent for oils, fats, resins, dyes, inks, waxes, platform molecule for the synthesis of key chemical compounds such as acetaldehyde, acetic acid, ethene, butadiene, elastomers, biofuel synthesis processes, etc.) [6,7,8], as well as food and drink industry (used as a natural product to extract and concentrate flavors and aromas, antimicrobial agent in pizza crust) [9]. A large amount of ethanol vapors present in the air may cause adverse central nervous system effects such as headache, mental excitement or depression, unconsciousness, and coma. Monitoring the level of ethanol is an essential process in various fields, such as environmental monitoring (air pollution caused by ethanol emissions) [10], food quality assessment (ethanol, along with CO_2_, are the primary spoilage metabolites of the freshly cut fruit) [11], traffic management (breath alcohol concentration is the routine test for Measurement of drunkenness level for drivers) [12,13], agriculture (measuring ethanol levels in crops) [14], medical field (breathalyzer) [15], alcoholic beverage industry [16], and ethanol production in biofuel plants [17]. Thereby, in recent decades, a lot of sensing principles and technologies such as conductometric [18], resistive [19,20], field effect transistor [21], FTIR, RAMAN, UV-VIS [22,23], surface acoustic wave [24], optical fiber [25], electrochemical [26], capacitive [27] have been developed or improved to manufacture ethanol sensor with superior performances.

Apart from the working principle of sensors and their design, the materials selected as the sensing layer appear as a key element in developing ethanol gas sensors with improved characteristics such as sensitivity, selectivity, response time, recovery time, hysteresis, and repeatability. Consequently, several materials have been explored as sensing layers within the design of ethanol sensors: metal–organic frameworks [28], metal oxide semiconductors [29], porous silicon [30], conducting polymers [31], metal sulfides [32], dielectric polymers [33], porphyrins [34], or SiC [35]. However, most of these sensors operate at high temperatures, leading to high energy consumption. Only a few studies have demonstrated metal oxide-based ethanol sensors functioning at room temperature.

Furthermore, many carbon-based materials are extensively used as sensing layers within the design of the ethanol gas sensors. Outstanding properties of these materials, such as large specific surface area, ability to interact with target molecule at room temperature, high mechanical strength, fast charge transfer, high stability, versatile covalent and noncovalent functionalization, and environmentally friendly have triggered substantial research to investigate their potential as ethanol gas sensing materials [36,37,38,39,40,41]. Carbon nanotubes [36], graphene oxide [37], reduced graphene oxide [38], carbon nanofiber [39], carbon nanodots [40], and graphene [41] are some of the carbonaceous nanomaterials studied for ethanol sensing applications. In recent years, carbon nanohorns (nanostructures composed of sp^2^-hybridized carbon atoms forming a conical shape with diameters of 2–5 nm and lengths of 30–50 nm [42]) have garnered growing attention for gas sensing applications. This interest is driven by their exceptional properties, including clean synthesis methods, the availability of high-purity samples, excellent chemical and thermal stability, large specific surface area, and low toxicity. Thus, both pristine and functionalized carbon nanohorns and their nanocomposites/nanohybrids were used as sensing layers within the design of resistive sensors for the detection and monitoring of relative humidity [43,44,45,46,47,48,49,50,51], ammonia [52], and ozone [53]. Moreover, holey carbon nanohorns (CNHox) were used as a sensing layer to detect and monitor ethanol gas [54,55]. In recent years, scientists have increasingly focused on designing room-temperature ethanol sensors due to their potential for real-time monitoring and low power consumption. This study aims to explore the room-temperature ethanol sensing properties of a novel nanohybrid, to discover new ways to reduce electric power consumption in next-generation, environmentally friendly sensors for Internet of Things (IoT) applications.

This paper presents the ethanol detection response of a resistive sensor based on a novel sensitive layer, a quaternary nanohybrid comprising CNHox, GO, SnO_2,_ and PVP at 1/1/1/1 *w*/*w*/*w*/*w* mass ratio. The proposed nanohybrid sensing layer for resistive detection of ethanol vapors includes four components whose chemical, physical, and electrical properties recommend them as key elements for the sensitive material. CNHox exhibits outstanding properties, such as high conductivity, high dispersibility, uniform size, excellent porosity, thermal and chemical stability, high adsorption capacity, superior permeability, exceptional catalytic properties, large specific surface area, low toxicity, and clean synthesis process (no metal catalyst is involved in their synthesis; thus, the produced CNHox are free of metal impurities) [55].

## 2. Materials and Methods

### 2.1. Materials

All the chemicals used in the sensing experiments were bought from Sigma Aldrich (Burlington, MA, USA). The chemicals used were of the highest available purity and were utilized without further modification. The metal-free CNHox powder (0% metallic compound) is characterized by a specific surface area of around 1300–1400 m^2^/g (according to the Brunauer–Emmett–Teller evaluation method) with lengths between 40 nm and 50 nm and diameters between 2 and 5 nm (Figure 1a).

According to the supplier, the amorphous graphite of CNHox cannot exceed 10%. PVP has an average molar weight of 29,000 Da (Figure 1b). CNHox powder is used as received. GO (4–10% edge-oxidized, average number of layers 15–20) is used as a dispersion in water, 2 mg/mL (Figure 1c). Tin (IV) oxide (SnO_2_) purchased as a powder (99.9% purity) has an average nanoparticle size lower than 100 nm and a specific surface area of around 10–25 m^2^/g 2-Propanol used in the synthesis is anhydrous (99.5%).

### 2.2. Sensing Layer Characterization

Raman spectra were recorded at room temperature using a Witec Raman spectrometer (Alpha-SNOM 300 S, WiTec GmbH, Ulm, Germany) with 532 nm excitation. A 532 nm diode-pumped solid-state laser, delivering 145 mW of power, was focused onto the sample with a 6 mm working distance objective on a Thorlabs MY100X-806 (Newton, NJ, USA) microscope, producing a laser spot size of ~1.0 µm. The spectra were collected in back-scattering geometry with 600 grooves/mm grating, using an exposure time of 20 s per accumulation. Calibration was performed with the 520 cm^−1^ Raman line of a silicon wafer; data acquisition and processing were conducted using WiTec Project Five software (WITec Project 5.1).

Scanning electron microscopy (SEM) was used to examine the surface topography of the sensing films. Surface visualization was performed using a field emission gun scanning electron microscope (FEG-SEM), Nova NanoSEM 630 (Thermo Scientific, Waltham, MA, USA), offering high-resolution imaging at low voltages and excellent surface sensitivity. The samples were analyzed without any preparation, and a measurement current of 1 nA was applied.

### 2.3. Preparation of the Sensing Layer

The synthesis of the sensing layer based on nanohybrid CNHox/GO/PVP/SnO_2_ = 1/1/1/1 (*w*/*w*/*w*/*w*) was conducted as described in the following section [56]. PVP solution was prepared by dissolving 6 mg of polymer in 10 mL of 2-propanol under stirring in the ultrasonic bath. Then, 6 mg of CNHox is added slowly to the PVP solution, under stirring. The suspension homogenization was achieved by employing a mild sonication bath at 42 kHz, with an output power of 70 W. Then, 3 mL of GO water dispersion was added to the prepared alcoholic PVP and CNHox suspension solution and stirred in the ultrasonic bath for 6 h at room temperature. In the last step of the dispersion preparation, 6 mg of SnO_2_ was added and stirred for 3 h in an ultrasonic bath at room temperature. An annealing process was performed for solvent evaporation by heating the prepared dispersion for 12 h at 100 °C in a vacuum oven under low pressure (1 mbar). This procedure yields a uniform distribution of the CNHox, GO carbon nanoparticles, and the metal oxide semiconductor (SnO_2_) nanoparticles in the hydrophilic PVP polymer network.

Microelectronic lift-off processes prepared the flexible polyimide substrate containing the metallic interdigitated structure; firstly, a photoresist layer was deposited, thermally treated, and patterned on the flexible substrate. Then, the Au/Cr layer was deposited on the entire substrate by electron gun evaporation, with chromium deposited first as an adherence layer for the Au layer. The Au/Cr layer adhesion test using tape confirmed that the Au/Cr layer adheres strongly to the polyimide substrate. Then, the photoresist and the metal deposited over it were removed by the well-known photoresist development process. Thus, the Au/Cr metal layer remained only in the desired layout regions, as shown in Figure 2, where the entire geometry of the IDT was present. The metal stripes of the Interdigitated Transducer (IDT) consisted of chromium with a thickness of 10 nm and gold with a thickness of 100 nm. The width and spacing of the digits were both 10 microns, and there is a separation of 0.6 mm between the digits and the bus bar. Finally, after masking the contact areas on the IDT structure from Figure 2, dispersing and sonicating the above final quaternary nanocomposite in isopropyl alcohol (IPA), the drop-casting method followed by 60 °C drying was used to generate the sensing film to be used in the ethanol detection experiments.

The ethanol chemiresistive sensor’s performance was evaluated by exposing it to varying ethanol concentrations in a 0.27 L testing chamber. To minimize moisture interference, after sealing the testing box, it was purged with nitrogen until the relative humidity (RH) was below 2%, as indicated by the commercial sensor SHT31 Sensirion (±2% RH accuracy). Ethanol was introduced in a controlled manner: a micropipette (0.1–2 µL range) dispensed volumes of 0.5–2.5 µL for low concentrations. For higher concentrations, a micro-syringe delivered 2.4 mg ethanol drops [56]. A magnetic stirrer (500 rpm for 8 min) ensured complete ethanol evaporation and homogeneous vapor distribution before measurements began. Ethanol drops were repeatedly weighed to minimize pipetting and syringe dispensing errors, with the mean drop weight used for calculations. Minimal measurement errors in three repeated cycles confirmed a strong correlation between the calculated ethanol concentration and the sensor’s resistance variations.

Before each test, nitrogen purging continued until the sensor’s resistance returned to its baseline value (R_i_ = 308 ohms), ensuring a zero relative variation in the measured resistance. The 8 min ethanol evaporation time was determined through preliminary tests to mitigate vaporization and diffusion errors, guaranteeing a stable resistance reading over time. Magnetic stirring was employed to prevent non-uniform ethanol distribution and adsorption onto chamber walls, as the latter contributed to sensor response variability at ethanol concentrations above 0.15 mg/cm^3^. All sensing experiments were conducted at room temperature in a temperature-controlled white chamber to prevent fluctuations. The high CNHox content (25% *w*/*w*) in the quaternary nanocomposite, well above the percolation threshold, ensured low and easily measurable electrical resistance, allowing for ultralow-power operation.

## 3. Results

The Raman spectra of the composite material CNHox/GO/SnO_2_/PVP reflect the individual contributions of these materials and provide insights into their interactions, structure, as well as modification in intensities due to the composite formation (Figure 3). PVP introduces new peaks, particularly in the C-H and C=O regions, as its Raman bands overlap with the bands of GO and CNHox.

The positions and analysis of the vibration modes of each material in the composite were summarized in Table 1 [57,58,59,60,61].

The discontinuous character of particles’ distribution on the sensitive layer’s surface can be observed in all recorded SEM (Figure 4). The nanoscale images show a particle size distribution in the range of 10–100 nm, similar to the size of synthesis precursors, proving an excellent dispersion process in the selected solvent. In addition, a very high porosity of the quaternary nanocomposite is revealed, supporting the high sensitivity of the sensing layer at room temperature.

SEM-EDX analysis results are shown in Figure 5, which reveals the elemental composition of the sensitive film (symbolized in both atomic and weight percentages). The method confirmed the presence of the elements C, Sn, O, N, and Si in agreement with the chemical composition of the synthesized nanohybrid. However, these quantitative evaluations may undergo errors specific to the EDX analysis due to the unavoidable inhomogeneous distribution of CNHox, GO, and SnO_2_ in the PVP network.

The sensing results are presented in Figure 6 and Figure 7, showing that the sensor’s response is reproducible for the three measurement cycles in the 0–0.16 mg/cm^3^ concentration range.

The transfer function of the sensor was presented by calculating the electrical resistance change relative to the initial resistance ((R_f_ − R_i_)/R_i_) as a function of ethanol concentration in the testing chamber (Figure 6 and Figure 7).

This makes the results dimensionless and more manageable to compare across different experiments, regardless of the absolute resistance values. Another reason for using relative change in the resistance is directly related to the sensitivity of the chemiresistive sensor. Using a relative change emphasizes how significant the response was compared to the sensor’s baseline performance. Uncontrollable factors, such as variations in fabrication, temperature, or environmental conditions, can influence absolute resistance values. The relative change minimizes the effect of variations induced during the fabrication process, with a focus on the dynamic response caused by the stimulus. At the same time, proper monitoring and control of the environment (temperature and relative humidity) was implemented throughout the experimental work.

In the representation of the relative variation in resistance as a function of ethanol concentration, three response zones of the sensor are highlighted (Figure 6). In the 0–0.088 mg/cm^3^ concentration range, the relative resistance variation in the sensor shows a quasi-linear increase, the slope of the trend line being lower than that associated with the linear response for the 0.088–0.16 mg/cm^3^ ethanol range (Figure 7).

Figure 6 presents the variation of (R_f_ − R_i_)/R_i_ with ethanol vapors concentration, where R_i_ is the resistance of the sensing film before exposure to ethanol, and R_f_ is the resistance after ethanol exposure. The chemiresistive sensing structure exhibits a good response to alcohol vapor concentrations varying from 0.008 to 0.16 mg/cm^3^, with resistance increasing over almost the entire range of ethanol concentrations, showing a maximum for ethanol concentration of 0.16 mg/cm^3^. From the transfer function as shown in Figure 7, it is evident that the sensor response is not linear, but it can be linearized for ethanol concentrations. Thus, one can observe that for low ethanol concentration (0.008 to 0.08 mg/cm^3^), the linearity is good (R^2^ = 0.88), with a slope of 1.606 (Figure 7), while for larger ethanol concentration (0.088 to 0.160 mg/cm^3^), the linearity is very good (R^2^ = 0.98), with a slope of 20.638 (Figure 7). The analysis of the experimental results also showed that the sensitivity for the range of high ethanol concentration (0.088 to 0.160 mg/cm^3^) was approximately 12 times higher than for low ethanol concentration (0.008 to 0.08 mg/cm^3^). The resistance of the tested sensor was measured after its readings were stabilized. In the concentration range of 0 to 0.089 mg/cm^3^, the stabilization time ranged from 80 to 120 s. However, when the ethanol concentration in the test chamber exceeded 0.089 g/cm^3^, the sensor’s equilibration time increased by 10 to 20 s. The performance of already tested ethanol sensors varies significantly. Some studies report response times ranging from a few seconds to tens of seconds, while others indicate that response times can decrease from tens of seconds to just a few seconds. Similarly, recovery times can drop from several hours to just a few minutes, depending on the sensor’s configuration, detection method, sensing layer, and operating temperature [62].

At concentrations higher than 0.16 mg/cm^3^ (inset III in Figure 6), the sensor’s response shows an increase in the relative variation in resistance value compared to the ethanol concentration. Still, the calculated values no longer follow a linear trend, and the response is not reproducible. Saturation effects are likely the primary reason for this result. Additionally, capillary condensation may influence the performance of the manufactured sensor at elevated ethanol concentrations.

## 4. Discussion

Considering the possible interactions between ethanol and the materials within the quaternary nanohybrid-based sensing layer employed, three distinct ethanol detection mechanisms can be identified and analyzed:

### 4.1. Ethyl Alcohol Acts as an Electron Donor for CNHox and GO

The first mechanism explaining the ethanol detection considers that CNHox and GO exhibit typical electrical behavior for p-type semiconductor materials (the movement of holes inside them causes the main current flow in such nanocarbon materials). At an interaction with CNHox and GO, ethanol molecules donate their electron pairs to these p-type semiconductor materials, which recombine with the holes from their valence band, and thus, the number of holes (majority carriers) is decreasing in both carbon structures. Accordingly, the sensitive layer becomes less conductive, which agrees with the experimental result from Figure 6 in the ethanol concentration range from 0 to 0.16 mg/cm^3^. Moreover, the interaction of ethanol molecules with CNHox and GO can be analyzed from the perspective of the hard–soft acid-base (HSAB) theory. Examples of hard, soft, and borderline acids and bases are given in Table 2. According to this theory, the chemical species react preferentially with similar hardness or softness species. As a result, hard bases tend to interact with hard acids, soft bases favor interactions with soft acids, and borderline acids generally bond with borderline bases. Like other chemical species containing oxygen atoms with lone electron pairs, ethanol molecules are classified as hard bases.

Table 2 shows carbocations (positively charged carbon ions) categorized as hard acids. Consequently, the holes within the structures of CNHox and GO can be regarded as hard acids. These holes neutralize each other through multiple recombination processes [63,64]. Finally, according to HSAB theory, Sn^4+^ ions are classified as hard acids, making a “hard acid–hard base” interaction between ethanol and SnO_2_ highly probable.

Thus, HSAB theory supports the feasibility of interactions between all components of the sensitive layer and ethanol, as well as the recombination of electrons and holes in the p-type semiconductor materials, CNHox and GO, aligning with the reasoning above based on the charge carrier recombination principle in semiconductors.

### 4.2. Electron Trapping and Generation from the SnO_2_-Oxygen-Ethyl Alcohol Interaction

SnO_2_ is a semiconducting metal oxide with n-type conductivity, meaning the electron concentration is much higher than the hole concentration for this material. Therefore, the electrical conductivity is mainly performed by electrons. It is well known that, at the SnO_2_ grain boundaries between different SnO_2_ nanoparticles, there are depletion regions due to interface defects trapping the electrons, and these depletion regions act as energy barriers in the electrical conduction process.

During the detection process, the following chemical processes occur as follows: (i) firstly, gaseous residual oxygen molecules are adsorbed on the surface of the exposed SnO_2_ nanoparticles; (ii) secondly, the electrons from the conduction band of the SnO_2_ nanoparticles are attracted by these adsorbed oxygen molecules; and then (iii) converted to oxygen anions (O_2_^−^) according to the set of reactions shown below:O_2_ (gas) → O_2_ (ads)(1)O_2_ (ads) + e^−^ → O_2_^−^ (ads)(2)O_2_^−^ (ads) + e^−^ → 2O^−^(3)O^−^ + e^−^ → O^2−^(4)

Therefore, according to Equations (2)–(4), the concentration of free electrons in SnO_2_ will decrease, expanding the depletion regions. Subsequently, (ii) the adsorbed ethanol as a reducing gas will remove the oxygen anions from the SnO_2_ surface by generating CO_2_ and H_2_O (Equations (5) and (6) from below), and finally, the electrons from these oxidation reactions return to the body of the SnO_2_ nanoparticles, and the Fermi level of SnO_2_ also returns to a near-initial state. Simultaneously, the potential energy barrier is lowered, and the depletion layer becomes thinner. As a result, electrons can move more easily between nanoparticles, decreasing the sensing structure’s resistance [65].CH_3_CH_2_OH (ads) + 6O^−^(ads) → 2CO_2_ (g) + 3H_2_O (g) + 6e^−^(5)CH_3_CH_2_OH (ads) + 6O_2_^−^ (ads) → 2CO_2_ (g) + 3H_2_O (g) + 12e^−^(6)

Based on this plausible explanation, the sensitive layer is expected to become more conductive due to the interaction between SnO_2_ nanoparticles and ethanol molecules. However, according to Figure 6, the experimental chemiresistive sensor response to the ethanol concentration increase shows a continuous rise in the electrical resistance with the concentration of ethanol in the range of 0–0.16 mg/cm^3^, which indicates that this mechanism is not prevailing, at least in this concentration range.

### 4.3. Swelling of PVP

According to Zereshki et al. [66], ethanol molecules have a strong interaction with PVP chains, and the dielectric polymer swells; the higher the ethanol concentration is, the higher the swelling is. The generation of the local water molecules, as described by Equations (5) and (6) above, may also contribute to the PVP swelling [67]. However, when the swelling process is developing, the distances between the conductive CNHox and low energy bandgap GO nanoparticles increase, and the number of electrical percolating pathways [68] decreases, as shown in Figure 8, and the overall electrical resistance increases.

According to this perspective, the sensitive layer is expected to become more resistive as the ethanol concentration steadily increases. The high sensitivity value in the ethanol concentration range of 0.08–0.16 mg/cm^3^ can be best explained by the increased hoping distance between these conductive segments, which may provide a much-increased value of the slope of the resistance dependence in Figure 6 and Figure 7. Therefore, the swelling phenomenon explains the experimental results in the ethanol concentration range of 0.08–0.16 mg/cm^3^. As we can see, when the ethanol concentration is higher than 0.16 mg/cm^3^, some regions where the sensitive layer becomes more conductive are identified.

Two alternative hypotheses can be taken into account. First, electron trapping and generation from the SnO_2_-oxygen-ethyl alcohol interaction (mechanism II, discussed above) may become prevalent. Secondly, the ionization of some water generated from reactions (5) and (6) may decrease the resistance of the sensing layer. Last but not least, capillary condensation can affect the behavior of the manufactured sensor at higher ethanol concentrations.

Besides these described mechanisms, the mutual interactions between nanohybrid constituents can be considered. Thus, π-π stacking interaction between GO and the nanohorns, as well as the hydrogen bonds between both nanocarbon materials and PVP, yield a possible supramolecular organization as presented in Figure 9.

It is also reasonable to assume the formation of islands of p-n semiconductor heterojunctions between graphene oxide, holey carbon nanohorns, and SnO_2_, which diminishes mutual interaction between organic constituents of nanohybrid. As a consequence, the specific surface area will be increased. Last but not least, mutual interaction between metal oxide semiconducting, yields changes in the pore distribution, which increases the specific surface area of nanohybrid exposed to ethanol vapors, affecting the number of active sensing sites. Considering that the resistance of the sensing layer increases across nearly the entire range of measured ethanol concentrations, it can be inferred that the p-type semiconductor behavior of both nanocarbon materials, along with the swelling of PVP, are the primary mechanisms responsible for the overall resistance of the sensitive layer when exposed to ethanol.

Each component of the quaternary nanohybrid used for resistive ethanol monitoring played a specific role. Unlike conventional ethanol sensors, which primarily rely on metal oxides and rare elements [69,70,71,72], CNHox and GO are derived from carbon-based materials recognized for their excellent electrical properties [73]. CNHox demonstrated exceptional properties, including increased conductivity (as a p-type semiconductor), high uniformity, a large surface area, easy synthesis (without metallic compounds), and sensitivity to alcohol molecules. These characteristics make CNHox a strong candidate for monitoring ethanol vapors at room temperature. GO offers several advantages, such as being a good charge carrier, enabling scalable fabrication, and serving as an effective dispersant for functionalized carbon nanohorns. Through intermolecular hydrogen bonding and π–π stacking interactions, GO can act as a dispersant for CNHox, helping to redisperse bundles of oxidized carbon nanohorns. Additionally, GO is a p-type material that shows reduced electrical conduction when exposed to ethanol. Polyvinylpyrrolidone (PVP) is an electrically insulating polymer with excellent binding properties and strong interactions with ethanol molecules.

A potential quantitative model to describe ethanol adsorption onto the sensing layer is the Freundlich adsorption isotherm. This model defines the relationship between the amount of gas adsorbed on a solid surface and the gas pressure. The Freundlich isotherm describes multilayer adsorption on a heterogeneous surface with varying binding energies, leading to the formation of multiple layers. Since adsorption sites have different affinities, the adsorption energy decreases as surface coverage increases. A key feature of the Freundlich model is the absence of a maximum adsorption capacity, meaning the adsorbent can continue adsorbing indefinitely, though at a diminishing rate [74].

Figure 9 illustrates the adsorption model. At ethanol concentrations below 0.08 mg/cm^3^, the ratio of [EtOH]_adsorbed_/m_adsorbent_ (x/m) increases rapidly due to the abundance of free adsorption sites. This results in a slow change in resistance, as only a small number of ethanol molecules are adsorbed at the interface, as shown in Figure 6 and Figure 7. When the ethanol concentration exceeds 0.08 mg/cm^3^, an inflection point appears in the graph showing the sensor’s measured response. This inflection point aligns with the one observed in the Freundlich isotherm. Beyond 0.08 mg/cm^3^, adsorption slows at the interface because the sensing layer surface becomes saturated with ethanol molecules. At this stage, the swelling of PVP accelerates, leading to a rapid increase in resistance as presented in Figure 6 and Figure 7.

This behavior demonstrates a clear correlation between the ethanol concentration and the Freundlich adsorption isotherm, where the rapid adsorption at lower concentrations and the subsequent saturation at higher concentrations align with the principles of the Freundlich model, explaining the dynamics of the sensor response dynamics observed in Figure 6 and Figure 7.

## 5. Conclusions

This study introduces a quaternary nanohybrid sensing layer (CNHox/GO/SnO_2_/PVP in a 1/1/1/1 mass ratio) designed explicitly for resistive ethanol vapor detection. Unlike conventional ethanol sensors, which primarily rely on metal oxides and rare elements, this innovative approach combines the synergistic properties of its components to enhance performance. CNHox provides high conductivity and porosity, improving electron transport and gas diffusion, while GO increases surface area and introduces functional groups that enhance ethanol interaction. SnO_2_ further strengthens ethanol adsorption and sensing response, and PVP ensures structural integrity and dispersion stability. Beyond performance, this sensor offers a cost-effective and environmentally friendly alternative to traditional designs. GO is a more affordable material than CNHox, helping to reduce production costs without compromising sensor quality, while SnO_2_ remains a low-cost yet effective complement to both GO and CNHox. By integrating these materials, the proposed sensor benefits from the high sensitivity of carbon-based nanomaterials while maintaining affordability and sustainability, making it a practical and scalable solution for ethanol-sensing applications. Last but not least, low power consumption (below 2 mW) makes these sensors promising alternatives in wireless sensor networks for Internet of Things applications, where energy constraint is one of the biggest challenges.

The sensing structure includes a flexible polyimide substrate and IDT-like electrodes. The sensing film used is a quaternary nanohybrid comprising two types of nanocarbon materials, CNHox and GO, a metal oxide semiconducting (SnO_2_) and a hydrophilic polymer (PVP), in a 1/1/1/1 mass ratio. The experimental setup was manufactured to investigate the response of the sensing layer deposited onto the substrate to various concentrations of ethanol vapors in a dry atmosphere. The developed resistive sensing structure showed good sensitivity to ethanol vapors over a broad range of concentrations (0.008–0.16 mg/cm^3^). In addition, the linearity of the sensor’s response was investigated at different ethanol concentration intervals, and the best sensing results were found in the range of ethanol concentrations between 0.088 and 0.160 mg/cm^3^. The analysis of the experimental sensing results emphasized that the sensitivity (i.e., the relative variation in the sensor resistance) for the range of high vapors ethanol concentration (0.088–0.16) mg/cm^3^ was approximately 12 times higher than for low analyte concentration (0.008 to 0.088 mg/cm^3^). Different mechanisms for explaining the ethanol vapors detection behavior of the developed sensing structure were discussed and assessed. Based on the obtained experimental data, it was argued that the p-type semiconductor behavior of CNOs and GO in conjunction with swelling of PVP are the dominant sensing mechanisms and yield an overall increase in the sensitive layer electrical resistance with ethanol vapor concentration. The HSAB principle is an additional solid argument that supports our interpretations regarding the sensing mechanism

## Figures and Tables

**Figure 1 sensors-25-01299-f001:**
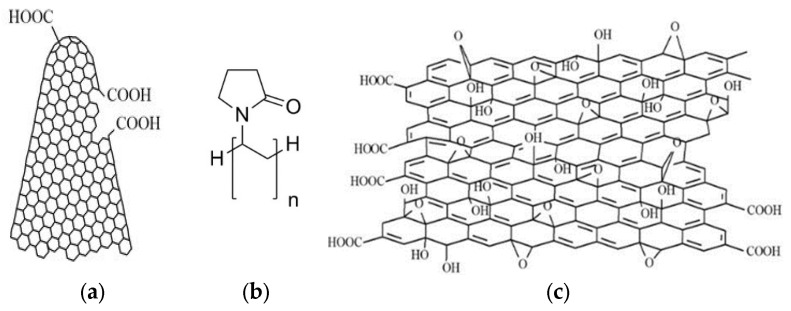
Structure of: (**a**) holey carbon nanohorns, (**b**) polyvinylpyrrolidone, and (**c**) graphene oxide.

**Figure 2 sensors-25-01299-f002:**
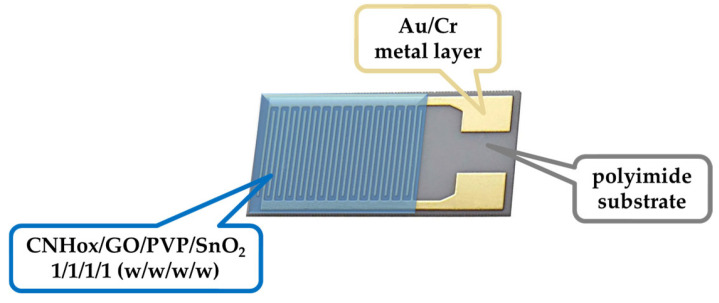
Layout of IDT sensing structure (chip area: 5 × 7 mm^2^).

**Figure 3 sensors-25-01299-f003:**
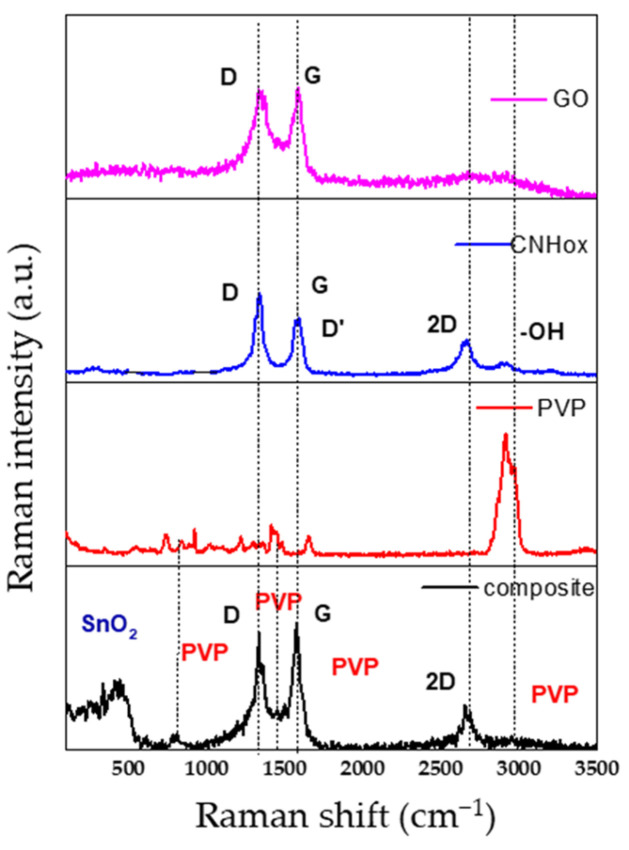
Raman spectra of the CNHox/GO/SnO_2_/PVP sensing layer deposited on a silicon substrate.

**Figure 4 sensors-25-01299-f004:**
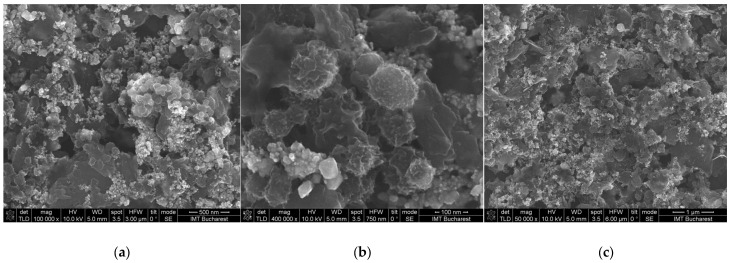
SEM images of the CNHox/GO/SnO_2_/PVP = 1/1/1/1 at 1:1:1:1 *w*/*w*/*w*/*w* ratio: (**a**) magnification 100,000; (**b**) magnification 400,000; (**c**) magnification 50,000.

**Figure 5 sensors-25-01299-f005:**
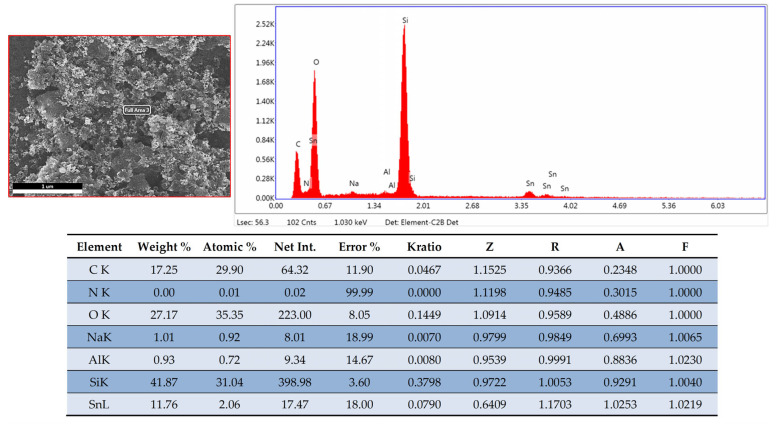
Surface composition of a quaternary nanohybrid comprising CNHox, GO, SnO_2_, and PVP (1:1:1:1 *w*/*w*/*w*/*w*) obtained by EDX spectroscopy coupled with SEM.

**Figure 6 sensors-25-01299-f006:**
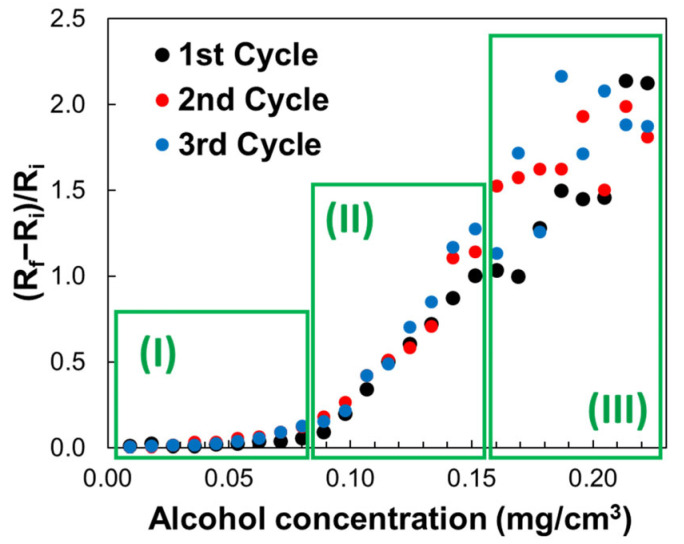
Graphical representation of the tested sensing device on three different cycles.

**Figure 7 sensors-25-01299-f007:**
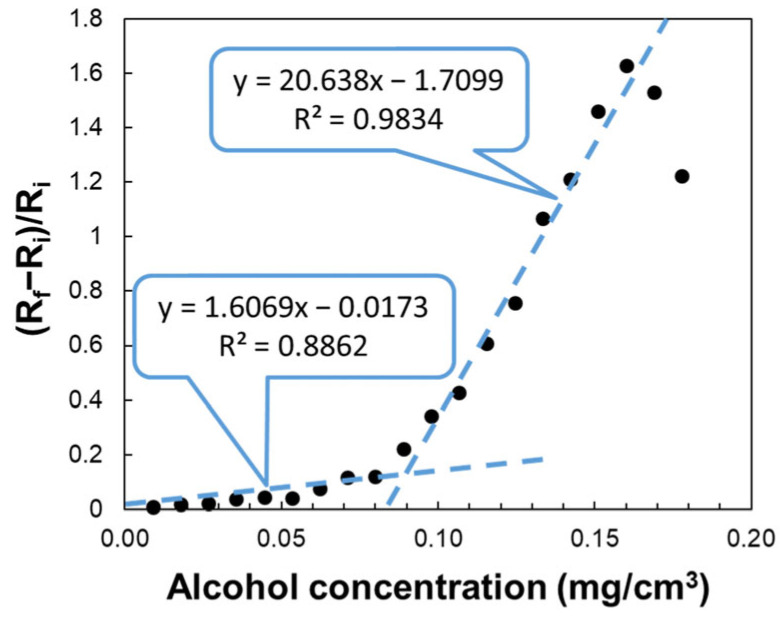
The measured response of the chemiresistive sensor to ethanol concentrations below 0.16 mg/cm^3^.

**Figure 8 sensors-25-01299-f008:**
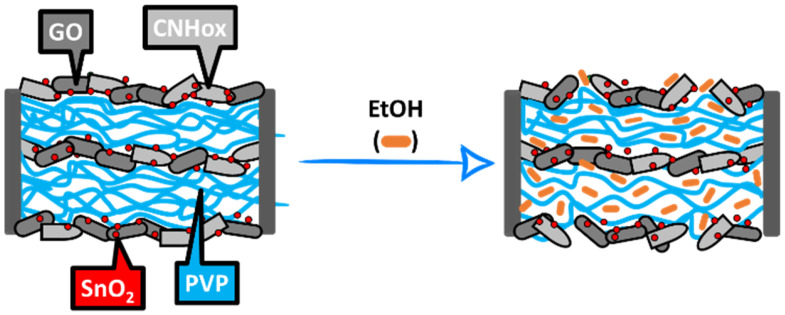
The swelling of PVP upon contact with ethanol molecules disrupts the percolating pathways of the CNHox and GO.

**Figure 9 sensors-25-01299-f009:**
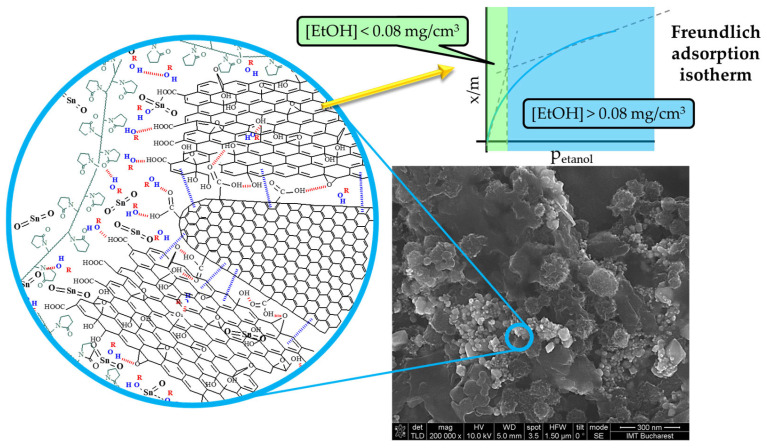
Possible architecture for supramolecular structure generated by CNHox, GO, SnO_2,_ and PVP.

**Table 1 sensors-25-01299-t001:** Analysis of the vibration modes of each material in the composite.

Material	Vibration Mode	Position (cm^−1^)	Analysis
CNHox [45,57]	D	~1350	Defects in the carbon network, oxygenated groups (epoxide, carboxyl)
	G	~1580	C-C vibrations in the carbon network (sp^2^ bond)
	2D	~2700	C-C stretching vibration in the graphite network
	D’	~1620	Defect vibrations, interactions with oxygenated groups
	OH (hydroxyl)	~3000–3200	Signal associated with -OH groups on nanohorns
GO [58,59]	D	~1350	Defects and discontinuities due to oxygenated groups
	G	~1580	C-C vibration in the carbon network
	2D	~2700	C-C stretching vibration
	D’	~1620	Structural defects and interactions with oxygenated groups
	OH (hydroxyl)	~3400	-OH groups on graphene oxide
	COOH (carboxyl)	~1700	C=O vibration of carboxyl groups
	Epoxid (C-O-C)	~1050	C-O vibration of epoxide groups
SnO_2_	A_1_g	~630	Symmetric Sn-O stretching vibration in the crystalline network
	B_2_g	~450	Asymmetric Sn-O bond vibration
	B_1_g	~750	Vibration around the tin atom in the tetragonal network
	E_1_g	~200	In-plane vibration of the network
	Lattice vibration	~120–180	Related to collective movements of oxygen and tin atoms throughout the entire crystalline network
PVP [60,61]	C-H	~1300	Stretching and deformation of C-H bonds in methylene groups
	C-C	~750–900~1350	Bending of C-C bonds in the polymer chain
	C-N	~1200–1250	Interactions between carbon and nitrogen atoms in the amide group
	C=O	~1650	C=O stretching vibration in the amide group
	C-H	~2800–3000	Stretching vibrations of C-H bonds in methylene (-CH_2_) and methyl (-CH_3_) groups

**Table 2 sensors-25-01299-t002:** Examples of hard, soft, and borderline acids and bases according to the HSAB theory.

	Hard	Borderline	Soft
Bases	C_2_H_5_OH, RO^−^, HO^−^, (CH_3_)_2_O, H_2_O, N_2_H_4_, R-NH_2_, CO_3_^2−^, F^−^, Cl^−^	C_6_H_5_NH_2_, C_5_H_5_N(pyridine), N_2_, Br^−^, N_3_^−^, Cl^−^	RSH, R_2_S, H_2_S, C_2_H_4_, CO, CN^−^, RCN, H_2_^−^, R_3_P, C_6_H_6_, RS^−^, I^−^
Acids	BCl_3_, SO_3_, H_3_O^+^, Li^+^, Mg^2+^, BF_3_, Al^3+^, Co^3+^, Sn^4+^, Ti^4+^, La^3+^, CR_3_^+^, In^3+^, Zr^4+^, CO_2_	Bi^3+^, Ni^2+^, Zn^2+^, Fe^2+^, Pb^2+^, Cu^2+^, Pb^2+^, BMe_3_	Cd^2+^, Cu^+^, Ag^+^, carbenes, I_2_, Hg^2+^, NO_2_, bulk metals

## Data Availability

The data that support the findings of this study are available from the corresponding author, M.B., upon reasonable request.

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
