# Peer review of "Holey Carbon Nanohorns-Based Nanohybrid as Sensing Layer for Resistive Ethanol Sensor"

_sensors, 2025, doi:10.3390/s25051299_

Round 1
Reviewer 1 Report
Comments and Suggestions for Authors
The paper presents a resistive ethanol sensor utilizing a quaternary nanohybrid comprising holey carbon nanohorns (CNHox), graphene oxide (GO), SnO₂, and polyvinylpyrrolidone (PVP). The sensor demonstrated good sensitivity to ethanol vapors within the concentration range of 0.008–0.16 mg/cm³. It emphasizes that ethanol molecules donate electrons to the carbon-based materials (CNHox and GO), affecting the overall conductivity of the sensor, with the swelling of PVP and electron trapping by SnO₂ also contributing to the response. The study also discusses the high sensitivity in the higher ethanol concentration range.
1. The paper should highlight more clearly the novelty and contribution of the research. Clearly emphasize unique aspects such as material selection, sensor design, and advantages over previous ethanol sensors.
2. Figures 6 and 7 should include a discussion on potential sources of error or uncertainties in the measurements. The paper would benefit from a discussion of the reproducibility of the sensor response, and how the sensor performs under varying environmental conditions, such as temperature and humidity.
3. Figures 3 and 5 both show obvious crop marks.
4. The interactions between the individual materials (CNHox, GO, SnO₂, and PVP) should be discussed in depth. This could include a more detailed analysis of their individual roles and how they contribute to the overall sensor functionality.
5. The manuscript refers to a mechanistic explanation of ethanol sensing. To verify these claims, previous work can be referenced. The following papers are suggested to be cited to supplement your research work, such as: Advanced Functional Materials, 2022., 32(9), 2109541; Nano Letters, 2023, 23(2), 637–644; Nano Research, 2024, 17(4), 2800–2807; Small Methods, 2024, DOI: 10.1002/smtd.202401485. A comparison with other similar sensor technologies or previous works in the field would help position this study more effectively. Currently, the uniqueness of the proposed sensor and its advantages over existing sensors are not clearly emphasized.
Author Response
Dear Reviewer thank you very much for your effort to read and analyze the submitted study. In the paragraphs presented below you will find a point-by-point response.
- The paper should highlight more clearly the novelty and contribution of the research. Clearly emphasize unique aspects such as material selection, sensor design, and advantages over previous ethanol sensors.
Response:
Thank you for the observation.
The following paragraph in the Conclusion section was added to emphasize the material selection and the advantages of the proposed sensing device.
“This study introduces a quaternary nanohybrid sensing layer (CNHox/GO/SnO₂/PVP in a 1/1/1/1 mass ratio), specifically designed for resistive ethanol vapor detection. Unlike conventional ethanol sensors, which primarily rely on metal oxides and rare elements, this innovative approach combines the synergistic properties of its components to enhance performance. CNHox provides high conductivity and porosity, improving electron transport and gas diffusion, while GO increases surface area and introduces functional groups that enhance ethanol interaction. SnO₂ further strengthens ethanol adsorption and sensing response, and PVP ensures structural integrity and dispersion stability. Beyond performance, this sensor offers a cost-effective and environmentally friendly alternative to traditional designs. GO is a more affordable material than CNHox, helping to reduce production costs without compromising sensor quality, while SnO₂ remains a low-cost yet effective complement to both GO and CNHox. By integrating these materials, the proposed sensor benefits from the high sensitivity of carbon-based nanomaterials while maintaining affordability and sustainability, making it a practical and scalable solution for ethanol-sensing applications. Last but not least, low power consumption (below 2 mW) makes these sensors promising alternatives in wireless sensor networks for Internet of Things applications, where energy constraint is one of the biggest challenges.”
- Figures 6 and 7 should include a discussion on potential sources of error or uncertainties in the measurements. The paper would benefit from a discussion of the reproducibility of the sensor response, and how the sensor performs under varying environmental conditions, such as temperature and humidity.
Thank you for your observation. Previous studies (Serban et al., 2020; Serban et al., 2021a; Serban et al., 2021b) have shown notable results using carbon nanohorns and their nanohybrids/nanocomposites as sensing layers in resistive sensor configurations. To ensure accurate detection of ethanol molecules and minimize interference from moisture, we maintain a low humidity level (less than 2%). Additionally, it is reasonable to expect a change in the resistance of the carbon nanohorn sensing layer with temperature variations. Carbon nanotubes, another type of nanocarbon material, and their nanocomposites are commonly used as temperature sensors. For this reason, the operating temperature was kept constant.
The following sources of errors were considered and the experimental setup was made to minimize them as presented below:
Measurement and handling errors:
- pipetting errors - slight variations in volume delivery can introduce errors, especially for very low ethanol concentrations.
- syringe dispensing errors -the weight of each ethanol drop may vary slightly, affecting concentration accuracy.
- weighing scale precision - if ethanol drop mass is measured, small inaccuracies in the scale could affect concentration calculations.
Environmental and systematic errors:
- nitrogen purging consistency - incomplete or inconsistent purging may leave residual moisture or residual ethanol between measuring cycles affecting ethanol adsorption and sensor response.
- temperature fluctuations - changes in room temperature can affect ethanol vapor pressure, influencing evaporation rate and sensor response
- humidity variations - residual moisture could interfere with ethanol adsorption and affect sensor readings.
- ethanol purity - impurities in the ethanol sample may introduce variations in sensor response. The ethanol used in the sensing experiments was bought from Sigma Aldrich, with a purity of 99.5 %.
Vaporization and diffusion errors:
- incomplete ethanol evaporation - ethanol may not fully vaporize leading to an uneven vapor concentration.
- non-uniform vapor distribution
- ethanol adsorption on box walls:
Sensor-related errors:
- baseline drift in sensor response - long-term use or environmental exposure can cause changes in the sensor's baseline resistance
- hysteresis effects -the sensor may retain residual ethanol molecules, leading to inconsistencies between successive measurements.
The presentation of the experimental setup was clarified to show the actions done to reduce the sources of errors in the measured result.
In the experimental section the paragraph that describes the experimental setup was modified as presented bellow:
“The ethanol chemiresistive sensor's performance was evaluated by exposing it to varying ethanol concentrations in a 0.27 L testing chamber. To minimize moisture interference, after sealing the testing box it was purged with nitrogen until the relative humidity (RH) was below 2% as indicated by the commercial sensor SHT31 Sensirion (± 2% RH accuracy). Ethanol was introduced in a controlled manner: for low concentrations, a micropipette (0.1–2 µL range) dispensed volumes of 0.5–2.5 µL. For higher concentrations, a micro-syringe delivered 2.4 mg ethanol drops [56]. A magnetic stirrer (500 rpm for 8 minutes) ensured complete ethanol evaporation and homogeneous vapor distribution before measurements began. Ethanol drops were repeatedly weighed to minimize pipetting and syringe dispensing errors, with the mean drop weight used for calculations. Minimal measurement errors in three repeated cycles confirmed a strong correlation between the calculated ethanol concentration and the sensor’s resistance variations.
Before each test, nitrogen purging continued until the sensor's resistance returned to its baseline value (Ri = 308 ohms), ensuring a zero relative variation of measured resistance. To mitigate vaporization and diffusion errors, the 8-minute ethanol evaporation time was determined through preliminary tests, ensuring a stable resistance reading over time. Magnetic stirring was employed to prevent non-uniform ethanol distribution and adsorption onto chamber walls, as the latter contributed to sensor response variability at ethanol concentrations above 0.15 mg/cm³. All sensing experiments were conducted at room temperature in a temperature-controlled white chamber to prevent fluctuations. The high CNHox content (25% w/w) in the quaternary nanocomposite, well above the percolation threshold, ensured low and easily measurable electrical resistance, allowing for ultralow-power operation.”
- Figures 3 and 5 both show obvious crop marks.
Thank you for your observation. The images have been replaced with graphs that meet the publisher's quality standards.
- The interactions between the individual materials (CNHox, GO, SnO₂, and PVP) should be discussed in depth. This could include a more detailed analysis of their individual roles and how they contribute to the overall sensor functionality.
Thank you for this observation. The following paragraph was added at the end of the discussion section to summarize the analysis of their individual roles and how they contribute to the overall sensor functionality.
“Each component of the quaternary nanohybrid used for resistive ethanol monitoring played a specific role. CNHox demonstrated exceptional properties, including increased conductivity (as a p-type semiconductor), high uniformity, a large surface area, easy synthesis (without metallic compounds), and sensitivity to alcohol molecules. These characteristics make CNHox a strong candidate for monitoring ethanol vapors at room temperature. GO offers several advantages, such as being a good charge carrier, enabling scalable fabrication, and serving as an effective dispersant for functionalized carbon nanohorns. Through intermolecular hydrogen bonding and π–π stacking interactions, GO can act as a dispersant for CNHox, helping to redisperse bundles of oxidized carbon nanohorns. Additionally, GO is a p-type material that shows reduced electrical conduction when exposed to ethanol. Polyvinylpyrrolidone (PVP) is an electrically insulating polymer with excellent binding properties and strong interactions with ethanol molecules.”
- The manuscript refers to a mechanistic explanation of ethanol sensing. To verify these claims, previous work can be referenced. The following papers are suggested to be cited to supplement your research work, such as: Advanced Functional Materials, 2022., 32(9), 2109541; Nano Letters, 2023, 23(2), 637–644; Nano Research, 2024, 17(4), 2800–2807; Small Methods, 2024, DOI: 10.1002/smtd.202401485. A comparison with other similar sensor technologies or previous works in the field would help position this study more effectively. Currently, the uniqueness of the proposed sensor and its advantages over existing sensors are not clearly emphasized.
The following references were added to emphasize the advantage o the proposed sensing device:
Li, Z., Liu, J., Yi, X., Wu, W., Li, F., Zhu, Z., ... & Liu, W. (2022). Metal–Organic Frameworks‐Based Fabry− Pérot Cavity Encapsulated TiO2 Nanoparticles for Selective Chemical Sensing. Advanced Functional Materials, 32(9), 2109541.
Li, Z., Liu, J., Feng, L., Pan, Y., Tang, J., Li, H., ... & Liu, W. (2023). Monolithic MOF-based metal–insulator–metal resonator for filtering and sensing. Nano Letters, 23(2), 637-644.
Liu, J., Feng, L., Li, Z., Wu, Y., Zhou, F., & Xu, Y. (2024). Plasma-etching on monolithic MOFs-based MIM filter boosted chemical sensing. Nano Research, 17(4), 2800-2807.
Li, Z., Tian, L., Wu, W., Feng, L., Khaniyev, B., Mukhametkarimov, Y., ... & Liu, J. (2024). Colorimetric Fabry‐Pérot Sensor with Hetero‐Structured Dielectric for Humidity Monitoring. Small Methods, 2401485.
Reviewer 2 Report
Comments and Suggestions for Authors
This paper investigates ethanol gas sensing mechanisms in a quaternary nanohybrid-based sensor comprising CNHox, GO, SnO₂, and PVP, analyzing the contributions of p-type semiconductor behavior, electron trapping in SnO₂, and PVP swelling. The findings suggest that the dominant sensing mechanism is the increase in resistance due to hole depletion in CNHox and GO, along with the swelling of PVP, which affects the conductive pathways. This article is well-written and contains useful information, but it is weak in expressing novelty.
Introduction:
I. Its important to mention clear what are the limitations of existing ethanol sensors that this study intends to overcome?
II. The introduction would be stronger if it clearly articulated the gap in the current literature and how this work advances the field.
III. Mentioning specific performance parameters (e.g., detection limit, response time, operating temperature) of existing ethanol sensors.
IV. Refer to new articles in field of carbon family-based sensor (DOI: 10.1016/j.colsurfa.2024.135563).
V. The introduction mentions various carbon-based materials used in gas sensing, but it would be useful to explicitly state what unique advantages holey carbon nanohorns (CNHox) provide over traditional carbon nanomaterials like CNTs or graphene for ethanol detection.
Materials and Methods:
VI. Specifying the actual purity percentages (e.g., 99.9%) for key materials would be necessary.
VII. Low pressure (1 mbar) is mentioned—was this done in a vacuum oven or another setup?
VIII. Mention the thickness of the Au/Cr layer.
IX. Was adhesion between Au/Cr and the polyimide substrate tested (e.g., via tape test)?
X. How was RH <2% verified?
XI. Was there a control experiment with just nitrogen to ensure no background effects?
Results:
XII. The discussion on Raman spectra mentions a decrease in the D band intensity due to SnO₂ but does not explicitly clarify whether this effect is due to chemical interaction or charge transfer effects.
XIII. The observation that responses at higher ethanol concentrations (>0.16 mg/cm³) are not reproducible is important. Consider discussing potential reasons for this, such as saturation effects or sensor degradation.
Discussion:
XIV. The text states that electron trapping in SnO₂ is not the prevailing mechanism at lower ethanol concentrations, but it is later mentioned as a possible mechanism at higher concentrations!
XV. Humidity is an important factor in the electrical conductivity of carbon family materials. (Look at this article: DOI: 10.1109/JSEN.2020.3038647). It is necessary to discuss the effect of humidity on electrical resistance in the paper or refer to the relevant reference in the manuscript.
XVI. Is there any quantitative model that can correlate swelling with resistance changes?
Author Response
Dear Reviewer, thank you for taking the time and effort to read and analyze our submitted study. Your observations have greatly helped us improve the quality of the study and highlight the novelty of the proposed sensing device. Below, you will find a point-by-point response to your detailed analysis.
- Its important to mention clear what are the limitations of existing ethanol sensors that this study intends to overcome?
Thank you for this observation. You are correct that commercial sensing devices measuring ethanol concentration have some limitations. To address this and to highlight the advantage of the proposed sensing device, the following paragraph has been added to the Introduction section (line 68):
“However, most of these sensors operate at high temperatures, leading to high energy consumption. Only a limited number of studies have demonstrated metal oxide-based ethanol sensors functioning at room temperature.”
2. The introduction would be stronger if it clearly articulated the gap in the current literature and how this work advances the field.
Thank you. The paragraph was added in the Introduction section:
“In recent years, scientists have increasingly focused on designing room-temperature ethanol sensors due to their potential for real-time monitoring and low power consumption. This study aims to explore the room-temperature ethanol sensing properties of a novel nanohybrid, with the goal of discovering new ways to reduce electric power consumption in next-generation, environmentally friendly sensors for Internet of Things (IoT) applications.”
3. Mentioning specific performance parameters (e.g., detection limit, response time, operating temperature) of existing ethanol sensors.
Thank you for the observation. A paragraph was added in the manuscript in the Results section:
“The performance of already tested ethanol sensors varies significantly. Some studies report response times in the range of a few seconds to tens of seconds, while others indicate that response times can decrease from tens of seconds to just a few seconds. Similarly, recovery times can drop from several hours to just a few minutes, depending on the sensor's configuration, detection method, sensing layer, and operating temperature.”
4. Refer to new articles in field of carbon family-based sensor (DOI: 10.1016/j.colsurfa.2024.135563).
The manuscript has been updated to reference sensors made from carbon-based materials, which are effective for detecting humidity, ammonia, alcohols, and more.
5. The introduction mentions various carbon-based materials used in gas sensing, but it would be useful to explicitly state what unique advantages holey carbon nanohorns (CNHox) provide over traditional carbon nanomaterials like CNTs or graphene for ethanol detection.
The following paragraph was added in the Introduction section
“CNHoxs exhibit outstanding properties, such as high conductivity, high dispersibility, uniform size, excellent porosity, thermal and chemical stability, high adsorption capacity, superior permeability, exceptional catalytic properties, large specific surface area, low toxicity, clean synthesis process (no metal catalyst is involved in their synthesis; thus, the produced CNHoxs are free of metal impurities).”
Materials and Methods:
6. Specifying the actual purity percentages (e.g., 99.9%) for key materials would be necessary.
The manuscript was modified as requested. Additional information about the key materials was presented in the Materials and Methods section.
7. Low pressure (1 mbar) is mentioned—was this done in a vacuum oven or another setup?
The annealing was done in a vacuum oven. Information was added in the manuscript. Thank you.
8. Mention the thickness of the Au/Cr layer.
The information was added in the Materials and Methods section.
The metal stripes of the Interdigitated Transducer (IDT) consist of chromium with a thickness of 10 nm and gold with a thickness of 100 nm. The width and spacing of the digits are both 10 microns, and there is a separation of 0.6 mm between the digits and the bus bar.
9.Was adhesion between Au/Cr and the polyimide substrate tested (e.g., via tape test)?
Thank you for your observation. The following information has been added to the manuscript:
"The Au/Cr layer adhesion test using tape confirmed that the Au/Cr layer adheres strongly to the polyimide substrate."
10. How was RH <2% verified?
A Sensirion commercial sensor for RH measurement (accuracy ±2%) was placed in the testing chamber to monitor the relative humidity. The information was added in the Experimental setup description.
1. Was there a control experiment with just nitrogen to ensure no background effects?
Yes, nitrogen gas does not affect the resistance of the sensing layer. Being an inert molecule, nitrogen does not impact the sensor's response. Purging the testing chamber with nitrogen allows the sensor to recover, and the sensor's resistance remains unchanged regardless of the duration of nitrogen purging. This information has been included in the manuscript.
Results:
12. The discussion on Raman spectra mentions a decrease in the D band intensity due to SnO₂ but does not explicitly clarify whether this effect is due to chemical interaction or charge transfer effects.
The information was added in the manuscript:
“One possible explanation is that the p-n heterojunctions formed by CNHox and GO, which act as p-type semiconductors on one side, along with SnO2 functioning as an n-type semiconductor on the other side, contribute to a reduction in the intensity of the D band.”
13. The observation that responses at higher ethanol concentrations (>0.16 mg/cm³) are not reproducible is important. Consider discussing potential reasons for this, such as saturation effects or sensor degradation.
The paragraph was added in the manuscript:
Saturation effects are likely the primary reason for this result. Additionally, capillary condensation may influence the performance of the manufactured sensor at elevated ethanol concentrations.
Discussion:
14. The text states that electron trapping in SnO₂ is not the prevailing mechanism at lower ethanol concentrations, but it is later mentioned as a possible mechanism at higher concentrations!
Thank you for your observation. Indeed, at higher concentrations of ethanol, we experienced a change in the prevalent mechanism.
15. Humidity is an important factor in the electrical conductivity of carbon family materials. (Look at this article: DOI: 10.1109/JSEN.2020.3038647). It is necessary to discuss the effect of humidity on electrical resistance in the paper or refer to the relevant reference in the manuscript.
Some previous studies( Serban, B. C., Buiu, O., Dumbravescu, N., Cobianu, C., Avramescu, V., Brezeanu, M., ... & Nicolescu, C. M. (2020). Oxidized Carbon Nanohorns as Novel Sensing Layer for Resistive Humidity Sensor. Acta Chimica Slovenica, 67(2); Serban, B. C., Buiu, O., Dumbravescu, N., Cobianu, C., Avramescu, V., Brezeanu, M., ... & Nicolescu, C. M. (2021). Oxidized carbon nanohorn-hydrophilic polymer nanocomposite as the resistive sensing layer for relative humidity. Analytical Letters, 54(3), 527-540. Serban, B. C., Cobianu, C., Buiu, O., Bumbac, M., Dumbravescu, N., Avramescu, V., ... & Radulescu, C. (2021). Ternary nanocomposites based on oxidized carbon nanohorns as sensing layers for room temperature resistive humidity sensing. Materials, 14(11), 2705) used with notable results carbon nanohorns and their nanohybrids/ nanocomposites as sensing layers within the configurations of resistive sensors . Consequently, in order to have an accurate response towards ethanol molecules and avoid strong interference of moisture we try to maintain a low level of humidity (less than 2%).
Additional references were added to emphasize the effect of humidity on the sensing properties of the proposed device.
16. Is there any quantitative model that can correlate swelling with resistance changes?
The paragraph below and Figure 9 were added to correlate adsorption of ethanol, swelling and resistance changes.
"A potential quantitative model to describe ethanol adsorption onto the sensing layer is the Freundlich adsorption isotherm. This model defines the relationship between the amount of gas adsorbed on a solid surface and the gas pressure. The Freundlich isotherm describes multilayer adsorption on a heterogeneous surface with varying binding energies, leading to the formation of multiple layers. Since adsorption sites have different affinities, the adsorption energy decreases as surface coverage increases. A key feature of the Freundlich model is the absence of a maximum adsorption capacity, meaning the adsorbent can continue adsorbing indefinitely, though at a diminishing rate [74].
Figure 9 illustrates the adsorption model. At ethanol concentrations below 0.08 mg/cm³, the ratio of [EtOH]adsorbed/madsorbent (x/m) increases rapidly due to the abundance of free adsorption sites. This results in a slow change in resistance, as only a small number of ethanol molecules are adsorbed at the interface, as shown in Figures 6 and 7. When the ethanol concentration exceeds 0.08 mg/cm³, an inflection point appears in the graph showing the sensor's measured response. This inflection point aligns with the one observed in the Freundlich isotherm. Beyond 0.08 mg/cm³, adsorption slows at the interface because the sensing layer surface becomes saturated with ethanol molecules. At this stage, the swelling of PVP accelerates, leading to a rapid increase in resistance."
Thank you again for the thorough analysis of the presented study.
Best regards,
Authors
Round 2
Reviewer 1 Report
Comments and Suggestions for Authors
All of my concerns have be addressed
Author Response
Dear Reviewer,
Thank you very much for your time and for your help in improving the presentation of the study.
Best regards,
The Authors
Reviewer 2 Report
Comments and Suggestions for Authors
- Add numbers to the subheadings.
- Enhance the quality of Figure 5.
- Mention briefly how the Freundlich adsorption isotherm explains the relationship between ethanol concentration and sensor response.
Author Response
Dear Reviewer, thank you for the response. Below, you will find a point-by-point response to your detailed analysis.
- Add numbers to the subheadings.
Thank you. All subheadings are numbered in the resubmitted manuscript.
- Enhance the quality of Figure 5.
The quality of the Figure 5 was improved.
- Mention briefly how the Freundlich adsorption isotherm explains the relationship between ethanol concentration and sensor response.
The following phrase was added at the end of Discussion section:
“This behavior demonstrates a clear correlation between the ethanol concentration and the Freundlich adsorption isotherm, where the rapid adsorption at lower concentrations and the subsequent saturation at higher concentrations align with the principles of the Freundlich model, explaining the dynamics of the sensor response dynamics observed in Figures 6 and 7.”
Thank you
Best regards,
The Authors